# Mental Health Outcomes among Chinese College Students over a Decade

**DOI:** 10.3390/ijerph182312742

**Published:** 2021-12-03

**Authors:** Sibo Zhao, Jie Zhang, Lisu Peng, Wenhui Yang

**Affiliations:** 1Department of Sociology, Central University of Finance and Economics, Beijing 100081, China; sibozhao@cufe.edu.cn (S.Z.); lisupeng@email.cufe.edu.cn (L.P.); yangwenhui121@126.com (W.Y.); 2Department of Sociology, State University of New York at Buffalo State, 1300 Elmwood Avenue, Buffalo, NY 14222, USA

**Keywords:** college students, suicidal ideation, depression, optimism, self-esteem, China

## Abstract

Background: Economic growth in China has brought about significant social and psychological changes in society. Objective: This study aims to explore how the mental and psychological health of college students has changed over the past decade. Methods: We observed several cohort samples in a Chinese university over a decade and looked at five mental health outcomes, including suicidal ideation, depression, optimism, self-esteem, and perceived social support, throughout each year of testing. Results: Our study highlights the declining rates of suicidal ideation and depression, combined with relative stability and even small increases in optimism, self-esteem, and perceived social support across a range of demographic variables. Conclusions: The findings of this study imply that in the context of economic growth, stabilizing and improving positive mental health states can help prevent and reduce the risk of depression and suicidal ideation among college students. The study also highlighted the need for more public health campaigns and interventions in universities to help students cope with mental health problems.

## 1. Introduction

Since the 1980s, the Chinese economy has been growing at an unprecedented rate [1], with an average rate of 6–10%, suggesting rapid and major changes in many aspects of Chinese society. The effects of these changes on mental health, however, have been inconsistent in the literature. On one hand, economic expansion has been associated with better mental health across different countries [2,3]. For instance, using time series analysis from 1928 to 2007 [4] indicated that in the U.S., the overall suicide rate and the suicide rates of groups aged 25–34 years, 35–44, 45–54, and 55–64 years rose during economic contractions, and declined during economic expansions. In Portugal, researchers also found evidence of a negative correlation between the total number of suicides and the GDP growth rate, especially for male’s mortality [5]. Zhang and his colleagues [6] found that the decrease of suicide rates during 1982–2005 in China was significantly correlated with the rapid growth of the economy. On the other hand, unplanned urban growth and social inequality during periods of higher economic growth [7] may lead to higher levels of social anomie, and lower levels of social integration [8], subsequently increasing mental health issues [9].

Although it remains inconclusive whether the effect of economic conditions on mental health is significant across different cohorts, mental health problems in childhood and adolescence are certainly associated with wide and enduring economic impacts [10]. Thus, young people’s mental health is a serious and growing concern among school educators, family members, and mental health professionals. Academic researchers have particularly focused on negative aspects of mental health, such as suicidal ideation and depression [11]. A number of studies indicated that there have been large increases in both suicidal ideation and depression in terms of number and severity among college students [12,13]. Based on reports from the American College Health Association-National College Health Assessment (ACHA-NCHA), suicidal ideation rates in students increased from 9.6% in 2015 to 13.0% in 2018, whereas the rates of depression diagnoses have increased from 14.5% in 2015 to 18.4% in 2018 (2015, 2018).

However, trends regarding suicidal ideation and depression are mixed across different countries and regions over time [14,15]. For instance, Brener et al. found that suicidal ideation among high school students is significantly decreasing in the United States. Rates of suicide attempts, however, showed a significant increase from 1991 to 1997 [16]. In South Korea, official figures indicated a general trend of worsening mental health, with rising rates of suicide and depression in particular, and the lifetime prevalence of major depression rose from 3.1% in 2001 to 5.6% in 2006 [17]. Other studies, however, found no significant changes over years. For example, there was no significant decline in suicide rate in Australia during 1991–2000 [18]. Using meta-analyses, Costello et al. found no evidence of an increase in the rate of depression among British children and adolescents over the past 30 years [19]. Moreover, although a population-based study found that the change in Chinese suicide rates has significantly declined between 2002 and 2011 [20], other researchers drew their conclusions on psychopathology largely from one year of data [21,22]. Thus, whether there is a trend of declining in suicidal ideation and depression among Chinese college students is still unclear.

Beyond the negative aspects of mental health, researchers have also identified several positive psychological factors that indicate college students’ well-being, such as optimism and self-esteem [23,24]. Cross-sectional studies have generally found that optimism is associated with improved mental health by conferring resilience to stressful life events [25,26], developing effective coping strategies, improving self-rated health, and providing social support and confidence [27,28,29]. For example, one study found that optimists had more social support and greater use of positive reinterpretation adjustments in college, and, therefore, were prospectively associated with smaller increases in stress and depression, and greater increases in perceived social support [30].

Self-esteem refers to a person’s belief about self, and overall perceptions on self-competence and efficacy. A number of studies have revealed a negative relationship between self-esteem, suicidal ideation, and depression [31,32]. According to diathesis–stress models of depression, self-esteem plays an important role in buffering the effect of stressful events on suicidal ideation and depression [33]. The impact of depression on suicidality is also reduced when self-esteem increases [34]. Researchers argue that individuals with high self-esteem have more effective coping resources, and are, therefore, protected from experiencing depression and suicidal thoughts [35].

Perceived social support, as one of the most important protective factors in suicidal ideation and depression, has also received attention [36]. Using bootstrapping, Kleiman and Riskind [37] analyzed the data of 172 college students, and found that perceived social support buffered suicidal ideation through the utilization of social support, and increases in self-esteem.

Given the protective effects of optimism, self-esteem, and perceived social support against suicidal ideation and depression found in the current literature, we hypothesize that though changes in suicidal ideation and depression are decreasing over time, the development of the optimism, self-esteem, and perceived social support may be increasing, or at least reasonably stable. However, the overwhelming majority of previous findings are largely based on data from well-developed Western societies, and data in this respect from currently developing Asian countries, including China, are scarce. China’s higher education sector has been experiencing a significant transformation during the reform period, particularly after 1999 when the Chinese government made the decision to enlarge the scale of higher education. As a result, annual college enrollment increased from 1 million in 1998 to 7.9 million in 2018, according to Chinese census data [38]. Owing to the lack of trend data in China, few studies have examined how optimism, self-esteem, and perceived social support among Chinese college students have changed over time. Therefore, the main purpose of this study is to explore the changes of suicidal ideation, depression, optimism, self-esteem, and perceived social support for Chinese college students during 2007–2015, and investigate how mental health outcomes differ across different sociodemographic groups.

China has a large number of universities and university colleges, which are mostly publicly funded, and full-time enrollment is required for college students. Although the pressure from employment has led to increasing competition in the universities, one study showed that Chinese college students’ well-being actually increased from 2002 to 2017 [2]. Thus, we expect that in our study, suicidal ideation and depression decrease over time, but the level of optimism, self-esteem, and perceived social support increase, or at least remain relatively stable, over the same time period.

## 2. Methods

### 2.1. Participants

A survey-based questionnaire was distributed at a key university in Beijing, China. Each year since 2007, we have distributed an annual survey examining mental health and related factors among undergraduate students in this university in Beijing. Each year in October, approximately 80 undergraduate classes from 4 grades are randomly selected across all the departments (e.g., finance, accounting, management, and statistics) of the university. We then systematically sampled respondents in selected classes by approaching instructors for their consent, and scheduling a time for administering the questionnaire. Students in the selected classes answered the questionnaire in groups. Given the nature of the survey, it is likely that some respondents who answered the questionnaire the first year would retake the survey in the following two or three years, which could form an unbalanced repeated cross-sectional data set. To correct for this, we limited the retake rate to 2% so as to minimize sample selection bias over time. The current dataset, therefore, comprises pooled data sets with all available years, similar to previous studies [39].

The questionnaire included demographics, interpersonal and social interaction variables, as well as indicators of mental health. The study was approved by the Institutional Research Board at the university where the research was administered. Informed consent was obtained before each student answered the questionnaire, and all the respondents were informed of their right to refuse and stop the procedure any time they wished.

Owing to the omission of the suicidal ideation scale in 2016, this study only used data from 2007 to 2015. In total, 9956 students ranging from 17 to 24 years were recruited for this current study. Two hundred and forty questionnaires were omitted on the basis of the following three principles: sociodemographic data incomplete; more than three missing values among all items; and responses showing undulating curve or over-centralized, the latter two of which may lead to systematic errors in the measurement process, and affect the validity of this study [40]. This leaves a final total of 9716 valid observations. The final sample consisted of 3650 (36.7%) male and 6293 (63.3%) female students, which is consistent with the gender ratio of the university.

### 2.2. Instruments

*Suicidal ideation*. To assess the respondents’ suicidal ideation level, we used the scale developed in Kessler’s National Comorbidity Survey [41,42]. The scale of NCS consists of seven items, including thoughts, plans, gestures, and attempts associated with suicide. Responses of “yes” were coded as “1”, “no” responses were coded as “0.” The Chinese version of the NCS has been supported as an excellent measure of suicidal ideation in the general Chinese population [43].

*Depression.* To assess the respondents’ depression level, we used the Center for Epidemiologic Studies-Depression Scale (CES-D) [44]. CES-D covers affective, cognitive, behavioral, and somatic symptoms associated with depression. CES-D has 20 items, and for each of them, respondents rate their own feeling for the number of days in the past one week (0 = less than one day, 1 = 1 to 2 days, 2 = 3 to 4 days, and 3 = 5 to 7 days). The Chinese version of CES-D has been validated with a number of Chinese samples, and proved to be an excellent measure of depression in general Chinese populations [45]. The good internal consistency of the scale for this sample was confirmed with Cronbach’s alpha (α = 0.923).

*Self-esteem*. To assess the respondents’ self-esteem level, we used the self-report version of the Self-Esteem Scale (SES) [46]. The 10 items of the SES assess a person’s overall evaluation of his or her worthiness as a human being. Responses were coded on a 4-point scale ranging from 1 (strongly disagree) to 4 (strongly agree). We also computed separate scores for the five positively-worded and the five negatively-worded items of the SES. The Chinese version of SES has been validated with a number of Chinese samples, and proved to be an excellent measure of esteem in general Chinese populations [45]. The Cronbach’s alpha coefficient for this scale was α = 0.802.

*Optimism*. We assessed optimism by a single question: “Do you feel optimistic about the future?” This approach directly measures people’s expectation of good outcomes. The answer was categorized into: never (0); sometimes (1); and always (2). To assess for convergent validity, we included validated questionnaires that were both similar and dissimilar to the optimism assessment question. Consistent with hypothesized directions, there were significant positive and negative correlations with the SES (r = 0.368, *p* < 0.001) and the CES-D measures (r = −0.374, *p* < 0.001), respectively, which indicated a good validity of the optimism measurement.

*Perceived social support*. The 12-items Multi-dimensional Scale of Perceived Social Support (MSPSS) was used to assess respondents’ perceived social support from family, friends, and significant others [47]. Responses were rated on a 7-point scale ranging from 1 (very strongly disagree) to 7 (very strongly agree). The scores of each subscale ranged from 4 to 28, and the total score ranged from 12 to 84, where higher scores indicate a higher level of social support. The Chinese version of MSPSS has been confirmed as a high validity tool for estimating perceived social support, and can be successfully applied to the general Chinese population [45]. High internal consistency was found (Cronbach’s α = 0.940).

In considering the net effect of mental health, several demographic variables were included in the analyses. According to previous studies, sex, year in school, and hometown were significant predictors for psychopathology [48,49]. Thus, those three predictors were recoded into dummy variables, as female, freshman, and rural, respectively (Table 1).

### 2.3. Statistical Analysis

All data were analyzed using IBM SPSS Statistics, version 24.0. Analysis of variance were performed to evaluate statistical significance and heterogeneity. Effect sizes were estimated using Cohen’s *d* and partial η^2^. Multiple regressions were conducted to investigate the potential relationships between the study variables and suicide ideation in college students. All *p* values were two-tailed, and *p* values less than 0.05 was considered statistically significant.

## 3. Results

### 3.1. Description of the Data

The total sample size was 9716, and it was roughly evenly distributed across the nine-year period. From 2007 to 2015, suicidal ideation among Chinese college students showed a significant declining trend across different groups. During the study period, the mean score of suicidal ideation decreased from 0.25 to 0.13, and from 0.24 to 0.12 for male and female students, respectively. The suicidal ideation scores in freshmen and sophomore were the highest (0.19), whereas scores in senior students were the lowest (0.14). From 2007 to 2015, suicidal ideation showed an average downward trend across different grades. The mean score of suicidal ideation dropped from 0.25 to 0.12, and from 0.23 to 0.13 for urban and rural college students, respectively. The association between suicidal ideation and optimism, self-esteem, and social support were significantly negatively correlated (r = −0.182, *p* < 0.001; r = −0.199, *p* < 0.001; r = −0.113, *p* < 0.001, respectively). There was also a significant positive correlation between suicidal ideation and depression (r = 0.299, *p* < 0.001) (Table 2).

### 3.2. The Trend of the Suicidal Ideation and Depression

Figure 1 depicts how suicidal ideation has shown a significant downward trend in the past decade. In the past decade, the mean rate of suicidal ideation was between 0.13 (in 2015) and 0.24 (in 2007). No significant difference was found between male and female in changes of suicidal ideation, however, there were significant differences in suicidal ideation among students across each year (partial η^2^ = 0.007, *p* < 0.001), and across different grades (partial η^2^ = 0.003; *p* < 0.01) (Table 3).

The mean score of the depression for Chinese college students in the past decade declined from 16.32 to 12.46, with a small increase period during the year of 2011 and 2012 (Figure 1). We also found significant differences in depression among students across different years (partial η^2^ = 0.006, *p* < 0.001), and from different hometowns (partial η^2^ = 0.103, *p* < 0.001) (Table 3).

### 3.3. The Trend of the Optimism, Self-Esteem, and Perceived Social Support

The trend of the optimism among Chinese college students in the past decade suggested that optimism increased from 1.53 to 1.66 (Figure 2). Male students indicate greater rates of optimism than female students, and urban students rated higher in optimism than rural students (*p* < 0.001). There were significant differences in the optimism of students in different grades, and the scores in senior students were the highest among different grades (*p* < 0.001) (Table 4).

From 2007 to 2015, the score of self-esteem maintained a high relatively stable level (Figure 2). We found significant differences in the levels of self-esteem among students of different years, sex, grades, and hometown (*p* < 0.05) (Table 3).

Trends in perceived social support were relatively stable, with the exception of 2013 (Table 5).

We then used multiple regression analyses to examine how demographic factors and psychological variables influence suicidal ideation among college students (Table 6). In all models, the negative coefficients for eight years indicated that the trend of suicidal ideation declined over the past decade. This is consistent with the nine-year trend of suicidal ideation (Figure 1). The results showed no significant difference in suicidal ideation across sex and hometown (*p* > 0.05). However, seniors showed a significant decline in suicidal ideation compared to freshmen (*p* < 0.001). This finding is consistent with previous studies, which showed that students’ suicidal ideation showed a downward trend in the age range of 19–22 years old [50].

When depression was added as a variable in Model 3, we observed that students who reported high depression levels also had higher suicidal ideation rates than those students who reported low depression levels (B = 0.15, *p* < 0.001). After adding perceived social support, self-esteem, and optimism into the model, Model 7 showed that though depression was positively related to suicidal ideation, optimism, self-esteem, and perceived social support showed positive protective effects, which contribute to the downward trends in suicidal ideation.

## 4. Discussion

The main purpose of this study is to explore the changes of suicidal ideation, depression, optimism, self-esteem, and perceived social support for Chinese college students during 2007–2015. The findings indicated that suicidal ideation and depression in Chinese college students generally decreased from 2007 to 2015, whereas the level of optimism increased, and the trends of self-esteem and perceived social support are relatively stable over the same time period. Similar findings have also been documented in other studies with either national or provincial level data of China [20,51].

Some previous studies suggested that female students’ suicidal ideation was higher than that of male students [52,53]. However, the current study suggested that sex was not a strong predictor of suicidal ideation in the models, which is consistent with recent findings [54]. The findings can be attributed to the successful education of gender egalitarianism in China, and the characteristics of the college students’ sample in this study: for this sample of full-time college students aged between 17 and 24 years, a highly educated and young population in China, with limited experience of sexism in the workplace. Other populations rather than college students may have different results [55].

We also found that year in school was associated with decreased suicidal ideation, in which freshmen reported greater suicidal ideation than students in other grades. Entering university is a stressful experience for many students, as it involves a multitude of new academic, personal, and interpersonal challenges. First-year students must adapt to new approaches when teaching and learning in a new environment, as well as compete with peers. While they are establishing a new lifestyle and social network with new teachers and classmates, they are dealing with issues relating to their changing role in a new environment, as well as establishing one’s identity and personal values. With so many new experiences, feelings of hopelessness may arise, and have been identified as a common factor of suicide among rural Chinese samples [56]. However, as college students progress through grades, they are likely to become more familiar with their college life. In addition, perceived social support gained from daily life might also increase students’ abilities to cope with tough work demands, and thus, potential feelings of hopelessness may diminish, leading to reductions in suicidal ideation.

It is worth noting that rates of self-esteem and perceived social support were lowest in 2013. One reason for this could be that the proportion of students who chose “very high” and “high” significantly declined compared to other years, whereas students who chose “moderate” significantly increased in the year of 2013. Students also reported getting less support from friends in 2013 than in other years. Indeed, 89.7% of the students reported that they had fewer than five friends, but in other years, the proportion of people who had fewer than five friends ranged from 53.9% to 57.9%. In addition, students also perceived less support from school in 2013 than in other years. Moreover, about 16.4% of students indicated that they never communicated with teachers in a month, and 28.5% of students met with their teacher only once. The significant positive correlation between perceived social support and self-esteem could also be the reason why the decline in self-esteem and perceived social support are also evident in 2013.

Our results indicated that female students generally reported higher levels of perceived social support than male students. This is consistent with a previous study [57], and may be a reflection of the general tendency among female students to have a higher level of need for affiliation and social activity, leading to more positive social adjustment than male students. This is supported by research showing that female students are more concerned with achieving acceptance and good social interactions than male students are [58].

Our finding that male students generally rated higher in optimism and self-esteem than female students is an interesting one. Minitchell and Hirom found similar results in a sample of students aged 13 to 17 in the United Kingdom. They argued that, despite the great advances made by women in a variety of spheres, including higher levels of academic achievement, general self-esteem and belief in one’s self may still be lower than that of their male counterparts [59]. In China, market-oriented jobs have recently become more competitive, unstable, and freed from the expectation of egalitarian compensation. In this society, women are more likely to be laid off than men, and they are more likely to experience greater difficulty finding re-employment in the private sector [60]. It also continues to be a challenge for female university graduates to obtain a good job compared to their male counterparts, despite many female applicants being equally qualified. This sex discrimination and income gap in the job market, combined with more and more unmarried young women, including college students attempting to return to the domestic sphere as a ‘‘Good Wife, Wise Mother’’ [61], may contribute to why female college students score lower in optimism and self-esteem compared to male students. However, this gender difference needs to be substantiated by future research.

Klerman and Weissman argued that large changes over relatively short periods of time cannot be attributed to genetics [62]. Moreover, it is difficult to attribute causation in a correlation study [63]. Our investigation found that both suicidal ideation and depression showed downward trends among Chinese college students, suggesting that declines in this psychopathology coincide with increased optimism, relatively stable self-esteem, and perceived social support.

Consistent with past findings for suicidal ideation based on college student samples [64,65], our correlation results indicated that over a decade’s period, the sampled Chinese college students experienced a decline in suicidal ideation, whereas their optimistic feelings significantly went up over the decade.

According to Carver and Scheier [28], optimists expect the best for a wide range of reasons, from those due to internal factors (e.g., self-esteem), to those due to external factors (e.g., chance). Therefore, explanations of the change may be found in the rapid growth of the Chinese economy in the past 40 or so years. Along with increased income and improved daily life, Chinese people, including Chinese college students, became direct benefiters of the wealth, and, therefore, much more optimistic about their life and the future of their infrastructure. Increased optimism and life satisfaction reduced the strains that resulted from unmet aspiration, and, therefore, decreased the risk of depression, as well as suicidality [66].

## 5. Conclusions

In this study, we investigated how optimism, self-esteem, and perceived social support among Chinese college students have changed in the past decade. Our analysis suggested that suicidal ideation and depression are declining among Chinese college students, whereas optimism is increasing, and the trend of self-esteem and perceived social support are relatively stable. These findings demonstrate that during the past few years in China, stabilizing and improving positive mental health states can help prevent and reduce the risk of depression and suicidal ideation among college students. The study also highlighted the need for more public health campaigns, and interventions in universities using optimized prevention strategies to help students cope with mental health problems.

## 6. Limitations

The study contains several limitations. First of all, this investigation is limited in its inclusion of sociodemographic variables due to changes in the questionnaires used throughout the years. It is also possible that the change trend of the studied psychological variables may stem from sampling bias and response bias. Additionally, the current sample is not representative of the general college population, being more resourceful, high-achieving, and more urban. Our recruitment strategy thus limited the generalizability of our results. Last, the most conspicuous problem in this study is no direct measure of economic variables. As a result, the power of discussion related to the social context explanation, such as the effect of the growing economy on mental health, is seriously restricted. In future research, we should improve and enrich the questionnaire by adding more variables to improve our understanding of mental health among Chinese college students. Despite these limitations, our study highlights the declining rates of suicidal ideation and depression, combined with relative stability, and even small increases, in optimism, self-esteem, and perceived social support across a range of demographic variables.

## Figures and Tables

**Figure 1 ijerph-18-12742-f001:**
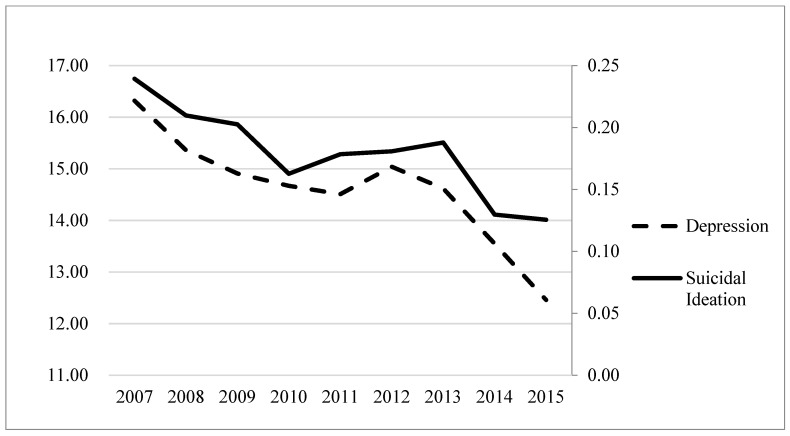
The trend of suicidal ideation and depression from 2007 to 2015.

**Figure 2 ijerph-18-12742-f002:**
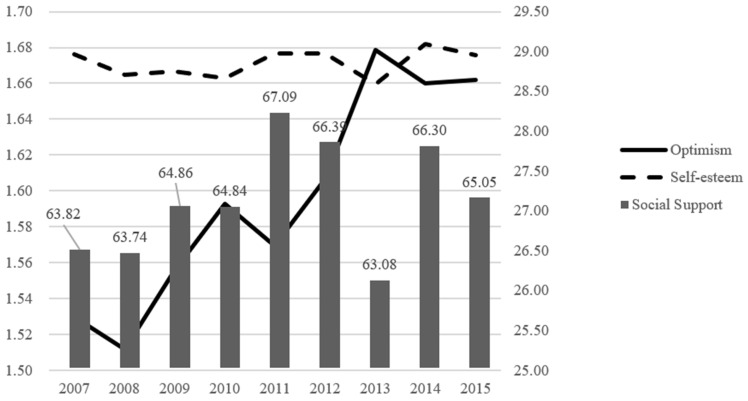
Trends in optimism, self-esteem, and perceived social support from 2007 to 2015.

**Table 1 ijerph-18-12742-t001:** Variable Descriptive Statistics (N = 9716).

Variable	f/Mean	%/S.D.	Value
Gender			0 = Female; 1 = Male
Female	6148	63.3%	
Male	3568	36.7%	
Year in school			1 = Freshman; 2 = Sophomore; 3 = Junior; 4 = Senior
Freshman	3004	30.9%	
Sophomore	2548	26.2%	
Junior	2265	23.3%	
Senior	1899	19.6%	
Hometown			0 = Rural; 1 = Urban
Rural	3902	40.2%	
Urban	5814	59.8%	
Suicidal ideation			0 = No; 1 = Yes
No	7985	82.18	
Yes	1731	17.82	
Depression	14.57	10.37	0 = Less than one day; 1 = 1–2 days; 2 = 3–4 days; 3 = 5–7 days
Optimism	1.60	0.51	0 = Never; 1 = Sometimes; 2 = Always
Self-esteem	28.87	3.87	1 = Strongly disagree; 2 = Disagree; 3 = Agree; 4 = Strongly agree
Perceived social support	65.26	12.35	1 = Very strongly disagree; 2 = Strongly disagree; 3 = Disagree; 4 = Uncertainty; 5 = Agree; 6 = Strongly agree; 7 = Very strongly agree

**Table 2 ijerph-18-12742-t002:** Mean rates of suicidal ideation, and correlations between suicidal ideation and optimism, self-esteem, depression, and perceived social support.

Year		Gender		Year in School				Hometown		Depression	Optimism	Self-Esteem	Perceived Social Support
		Male	Female	Freshman	Sophomore	Junior	Senior	Rural	Urban				
2007 (*n* = 961)	Mean/r	0.25	0.24	0.25	0.23	0.26	0.21	0.23	0.25	0.299 ***	−0.182 ***	−0.199 ***	−0.113 ***
	SD	0.43	0.43	0.43	0.42	0.44	0.41	0.42	0.43				
	*n*%	350 (36.4%)	611 (63.6%)	280 (29.1%)	273 (28.4%)	218 (22.7%)	190 (19.8%)	417 (43.4%)	544 (56.6%)				
2008 (*n* = 858)	Mean/r	0.22	0.20	0.19	0.23	0.22	0.20	0.19	0.22	0.234 ***	−0.120 ***	−0.121 ***	−0.072 *
	SD	0.42	0.40	0.39	0.42	0.41	0.40	0.39	0.42				
	*n*%	315 (36.7%)	543 (63.3%)	224 (26.2%)	238 (27.7%)	219 (25.5%)	177 (20.6%)	359 (41.8%)	499 (58.2%)				
2009 (*n* = 765)	Mean/r	0.19	0.21	0.24	0.24	0.18	0.13	0.21	0.20	0.228 ***	−0.116 ***	−0.170 ***	−0.083 *
	SD	0.39	0.41	0.43	0.43	0.38	0.33	0.41	0.40				
	*n*%	282 (36.9%)	483 (63.1%)	267 (34.9%)	154 (20.1%)	179 (23.4%)	165 (21.6%)	300 (39.2%)	465 (60.8%)				
2010 (*n* = 1051)	Mean/r	0.15	0.17	0.16	0.17	0.15	0.17	0.17	0.16	0.187 ***	−0.163 ***	−0.121 ***	0.006
	SD	0.35	0.38	0.37	0.38	0.36	0.38	0.38	0.37				
	*n*%	413 (39.3%)	638 (61.7%)	301 (28.6%)	300 (28.5%)	253 (24.1%)	197 (18.8%)	355 (33.8%)	696 (66.2%)				
2011 (*n* = 1877)	Mean/r	0.18	0.18	0.18	0.19	0.20	0.13	0.18	0.18	0.254 ***	−0.208 ***	−0.161 ***	−0.097 ***
	SD	0.39	0.38	0.39	0.39	0.40	0.34	0.39	0.38				
	*n*%	696 (37.1%)	1181 (62.9%)	525 (28.0%)	497 (26.5%)	484 (25.7%)	371 (19.8%)	721 (38.4%)	1156 (61.6%)				
2012 (*n* = 1222)	Mean/r	0.18	0.19	0.24	0.18	0.15	0.12	0.18	0.18	0.238 ***	−0.138 ***	−0.177 ***	−0.101 ***
	SD	0.38	0.39	0.43	0.39	0.36	0.33	0.38	0.39				
	*n*%	471 (38.5%)	751 (61.5%)	363 (29.7%)	346 (28.3%)	323 (26.4%)	190 (15.6%)	457 (37.4%)	765 (62.6%)				
2013 (*n* = 974)	Mean/r	0.17	0.20	0.21	0.21	0.17	0.13	0.21	0.17	0.215 ***	−0.180 ***	−0.142 ***	−0.116 ***
	SD	0.38	0.40	0.41	0.41	0.37	0.33	0.41	0.38				
	*n*%	341 (35.0%)	633 (65.0%)	321 (33.0%)	310 (31.8%)	192 (19.7%)	151 (15.5%)	390 (40.0%)	584 (60.0%)				
2014 (*n* = 917)	Mean/r	0.13	0.13	0.15	0.16	0.13	0.07	0.12	0.14	0.242 ***	−0.162 ***	−0.181 ***	−0.175 ***
	SD	0.34	0.33	0.35	0.37	0.34	0.26	0.33	0.34				
	*n*%	335 (36.5%)	582 (63.5%)	350 (38.2%)	177 (19.3%)	187 (20.4%)	203 (22.1%)	406 (44.3%)	511 (55.7%)				
2015 (*n* = 1091)	Mean/r	0.13	0.12	0.15	0.13	0.13	0.08	0.13	0.12	0.254 **	−0.163 ***	−0.164 ***	−0.203 ***
	SD	0.34	0.33	0.36	0.33	0.34	0.28	0.34	0.33				
	*n*%	365 (33.5%)	726 (66.5%)	373 (34.2%)	253 (23.2%)	210 (19.2%)	255 (23.4%)	497 (45.6%)	594 (54.4%)				
All (*n* = 9716)	Mean/r	0.18	0.18	0.19	0.19	0.18	0.14	0.18	0.18	0.246 ***	−0.169 ***	−0.160 ***	−0.107 ***
	SD	0.38	0.38	0.40	0.39	0.38	0.34	0.38	0.38				
	*n*%	3568 (36.7%)	6148 (63.3%)	3004 (30.9%)	2548 (26.2%)	2265 (23.4%)	1899 (19.5%)	3902 (40.2%)	5814 (59.8%)				

Note: * *p* < 0.05, ** *p* < 0.01, *** *p* < 0.001.

**Table 3 ijerph-18-12742-t003:** Suicidal ideation and depression in college students from 2007 to 2015.

Suicidal Ideation								Depression					
	Cohen’s d/η^2^	Mean	S.D.	Has the Subject Seriously Thought about Death?	Has the Subject Thought about Death in the Past 12 Months?	Has the Subject Made Plans about Suicide in the Past 12 Months?	Has the Subject Attempted Suicide in the Past 12 Months?		Cohen’s d/η^2^	Mean	S.D.	Low (≤16)	High (>16)
				N	N	N	N						
Year	0.007 ***							Year	0.006 ***				
2007		0.24	0.43	229	102	14	2	2007		16.32	10.43	56.9%	43.1%
2008		0.21	0.41	180	80	10	2	2008		15.36	10.26	60.1%	39.9%
2009		0.20	0.4	155	69	7	0	2009		14.91	9.92	61.8%	38.2%
2010		0.16	0.37	171	72	5	1	2010		14.67	10.55	62.2%	37.8%
2011		0.18	0.38	335	149	26	9	2011		14.51	10.46	63.1%	36.9%
2012		0.18	0.39	221	110	13	9	2012		15.04	10.17	62.4%	37.6%
2013		0.19	0.39	172	70	15	9	2013		14.62	10.52	62.9%	37.1%
2014		0.13	0.34	110	51	16	8	2014		13.55	10.47	67.6%	32.4%
2015		0.13	0.33	122	48	6	8	2015		12.46	9.97	70.9%	29.1%
Sex	0.001							Sex	0.023 *			0.0%	0.0%
Male		0.18	0.38	614	307	45	26	Male		14.42	10.71	64.2%	35.8%
Female		0.18	0.38	1081	444	67	22	Female		14.66	10.17	62.7%	37.3%
Year in school	0.003 **							Year in school	0.001			0.0%	0.0%
Freshman		0.19	0.4	568	246	32	14	Freshman		14.12	10.32	64.9%	35.1%
Sophomore		0.19	0.39	478	199	28	12	Sophomore		15.09	10.33	60.6%	39.4%
Junior		0.18	0.38	395	176	33	15	Junior		14.97	10.41	62.8%	37.2%
Senior		0.14	0.34	254	130	19	7	Senior		14.1	10.41	64.6%	35.4%
Hometown	0.001							Hometown	0.103 ***			0.0%	0.0%
Urban		0.18	0.38	1018	452	76	32	Urban		14.14	10.35	64.9%	35.1%
Rural		0.18	0.38	677	299	36	16	Rural		15.21	10.37	60.8%	39.2%

Note: * *p* < 0.05, ** *p* < 0.01, *** *p* < 0.001.

**Table 4 ijerph-18-12742-t004:** Optimism and self-esteem in college students from 2007 to 2015.

Optimism							Self-Esteem							
	Cohen’s d/η^2^	Mean	S.D.	Never	Sometimes	Always		Cohen’s d/η^2^	Mean	S.D.	Very High (35–40)	High (30–34)	Moderate (20–29)	Low (≤19)
Year	0.011 ***						Year	0.002						
2007		1.53	0.53	1.8%	43.5%	54.7%	2007		28.97	3.87	8.1%	32.0%	58.3%	1.6%
2008		1.51	0.53	1.5%	45.8%	52.7%	2008		28.71	3.87	8.9%	28.2%	62.0%	0.9%
2009		1.56	0.51	0.7%	43.1%	56.2%	2009		28.75	3.95	8.8%	27.7%	61.8%	1.7%
2010		1.59	0.53	1.8%	37.1%	61.1%	2010		28.67	3.6	6.2%	27.2%	65.6%	1.0%
2011		1.57	0.52	1.3%	40.5%	58.2%	2011		28.97	3.92	8.8%	32.3%	57.5%	1.4%
2012		1.61	0.52	1.8%	35.8%	62.4%	2012		28.98	3.74	7.9%	32.3%	59.4%	0.4%
2013		1.68	0.48	0.8%	30.5%	68.7%	2013		28.59	3.81	7.9%	27.0%	64.3%	0.8%
2014		1.66	0.49	0.9%	32.3%	66.8%	2014		29.1	4.05	10.4%	31.2%	57.3%	1.2%
2015		1.66	0.49	0.6%	32.5%	66.8%	2015		28.96	3.94	10.3%	27.4%	61.7%	0.6%
Gender	0.078 ***						Gender	0.052 *						
Male		1.62	0.52	1.9%	33.9%	64.2%	Male		29	3.96	9.7%	30.7%	58.3%	1.3%
Female		1.58	0.51	0.9%	40.1%	59.0%	Female		28.8	3.81	7.9%	29.3%	61.9%	0.9%
Year in school	0.003 **						Year in school	0.003						
Freshman		1.59	0.52	1.2%	39.1%	59.7%	Freshman		29	4	9.9%	29.8%	59.1%	1.2%
Sophomore		1.58	0.52	1.2%	39.1%	59.7%	Sophomore		28.59	3.72	6.8%	28.3%	63.9%	1.1%
Junior		1.57	0.53	1.6%	39.2%	59.1%	Junior		28.75	3.74	7.5%	29.8%	61.8%	0.9%
Senior		1.66	0.5	1.1%	32.3%	66.6%	Senior		29.18	3.96	10.0%	31.9%	57.1%	1.1%
Hometown	0.116 ***						Hometown	0.183 ***						
Urban		1.62	0.51	1.2%	35.6%	63.2%	Urban		29.15	3.94	9.8%	32.0%	57.1%	1.1%
Rural		1.56	0.52	1.4%	41.2%	57.4%	Rural		28.45	3.71	6.7%	26.6%	65.7%	1.1%

Note: * *p* < 0.05, ** *p* < 0.01, *** *p* < 0.001.

**Table 5 ijerph-18-12742-t005:** Perceived social support among college students from 2007 to 2015.

	Cohen’s d/η^2^	Mean	S.D.	High	Low
				(≥50)	(≤49)
Year	0.012 ***				
2007		63.82	10.53	91.4%	8.6%
2008		63.74	11.51	90.7%	9.3%
2009		64.86	11.68	90.7%	9.3%
2010		64.84	11.76	90.5%	9.5%
2011		67.09	12.17	92.8%	7.2%
2012		66.39	13.06	91.3%	8.7%
2013		63.08	14.05	86.8%	13.2%
2014		66.3	13.28	90.3%	9.7%
2015		65.05	11.94	91.3%	8.7%
Gender	0.223 ***				
Male		63.49	13.42	86.9%	13.1%
Female		66.29	11.56	93.1%	6.9%
Year in school	0.002				
Freshman		65.86	12.1	91.8%	8.2%
Sophomore		64.96	12.36	90.7%	9.3%
Junior		64.63	12.43	89.8%	10.2%
Senior		65.45	12.59	90.8%	9.2%
Hometown	0.195 ***				
Urban		66.22	12.45	91.6%	8.4%
Rural		63.83	12.06	89.7%	10.3%

Note: *** *p* < 0.001.

**Table 6 ijerph-18-12742-t006:** Multiple regression models for suicidal ideation.

Variables	Model 1	Model 2	Model 3	Model 4	Model 5	Model 6	Model 7
	*B* (S.E.)	*B* (S.E.)	*B* (S.E.)	*B* (S.E.)	*B* (S.E.)	*B* (S.E.)	*B* (S.E.)
Year							
2007	-						
2008	−0.030 (0.018)	−0.028 (0.018)	−0.024 (0.018)	−0.029 (0.018)	−0.033 (0.018)	−0.030 (0.018)	−0.028 (0.017)
2009	−0.037* (0.018)	−0.036 (0.018)	−0.029 (0.018)	−0.037 * (0.018)	−0.040 * (0.018)	−0.033 (0.018)	−0.032 (0.018)
2010	−0.077 *** (0.017)	−0.077 *** (0.017)	−0.069 *** (0.017)	−0.078 *** (0.017)	−0.083 *** (0.017)	−0.070 *** (0.017)	−0.072 *** (0.017)
2011	−0.061 *** (0.015)	−0.060 *** (0.015)	−0.051 ** (0.015)	−0.059 *** (0.015)	−0.061 *** (0.015)	−0.056 *** (0.015)	−0.052 *** (0.015)
2012	−0.058 *** (0.016)	−0.060 *** (0.016)	−0.052 ** (0.016)	−0.060 *** (0.016)	−0.060 *** (0.016)	−0.052 ** (0.016)	−0.051 ** (0.016)
2013	−0.051 *** (0.017)	−0.054 ** (0.017)	−0.045 ** (0.017)	−0.060 *** (0.017)	−0.060 *** (0.017)	−0.036 * (0.017)	−0.044 ** (0.017)
2014	−0.110 *** (0.018)	−0.109 *** (0.018)	−0.094 *** (0.017)	−0.111 *** (0.018)	−0.108 *** (0.017)	−0.094 *** (0.017)	−0.091 *** (0.017)
2015	−0.114 *** (0.017)	−0.113 *** (0.017)	−0.092 *** (0.017)	−0.113 *** (0.017)	−0.113 *** (0.017)	−0.097 *** (0.017)	−0.091 *** (0.016)
Gender							
Male	-						
Female		−0.003 (0.008)	0.001 (0.008)	−0.011 (0.008)	0.001 (0.008)	0.001 (0.008)	−0.001 (0.008)
Year in school							
Freshman	-						
Sophomore		−0.008 (0.010)	−0.013 (0.010)	−0.009 (0.010)	−0.014 (0.010)	−0.007 (0.010)	−0.014 (0.010)
Junior		−0.020 (0.011)	−0.022 * (0.010)	−0.023 * (0.011)	−0.024 * (0.011)	−0.020 (0.011)	−0.025 * (0.011)
Senior		−0.058 *** (0.011)	−0.059 *** (0.011)	−0.060 *** (0.011)	−0.056 *** (0.011)	−0.050 *** (0.011)	−0.054 *** (0.011)
Hometown							
Rural	-						
Urban		−0.001 (0.008)	0.005 (0.008)	0.001 (0.008)	0.010 (0.008)	0.006 (0.008)	0.013 (0.008)
Depression							
Low	-						
High			0.150 *** (0.008)				0.096 *** (0.009)
Perceived social support							
Low	-						
High				−0.119 *** (0.013)			−0.048 *** (0.014)
Self-esteem					−0.016 *** (0.001)		−0.007 *** (0.001)
Optimism	-						
Low							
Moderate						−0.206 *** (0.035)	−0.155 *** (0. 034)
High						−0.320 *** (0.034)	−0.212 *** (0.035)
Constant	0.239 *** (0.012)	0.259 *** (0.015)	0.193 *** (0.015)	0.371 *** (0.019)	0.711 *** (0.032)	0.517 *** (0.037)	0.638 (0.048)
Adjust R2	0.007	0.009	0.044	0.017	0.034	0.035	0.060
F	8.961	7.843	33.081	12.925	25.459	24.790	35.419
N	9716	9716	9716	9716	9716	9716	9716

Note: * *p* < 0.05, ** *p* < 0.01, *** *p* < 0.001.

## Data Availability

The data are available on reasonable request from the corresponding author.

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
