# Peer review of "Mental Health Outcomes among Chinese College Students over a Decade"

_ijerph, 2021, doi:10.3390/ijerph182312742_

Round 1

Reviewer 1 Report

The current cohort study examined how the recent rapid economic growth affecting the mental and psychological health of college students in China. Authors have looked at five mental health outcomes including suicidal ideation, depression, optimism, self-esteem, and perceived social support in almost 10,000 students in a University over a decade. The findings revealed declining rates of suicidal ideation and depression, stability in optimism, self- esteem, and perceived social support and ultimately improving positive mental health states of students over a decade. The significance of the findings is that it will fill the gap in knowledge and highlighted the need for more public health campaigns and interventions in universities to help students cope with mental health problems. The study is of good quality as authors have taken a meticulous approach to each aspect of the study:  design, methodology, analysis and discussion.  The authors had conducted a comprehensive literature search and discussion includes the strengths, limitations and implications of the study and findings in detail.

I have few comments to make:

  1. Survey-based questionnaires have high risk of introducing recall bias
  2. Generalizability is limited as the study is based at only one university ( urban, more resourceful, high-achiever students)
  3. More female students, than males were included. Did author looked at the any differential characteristics of non-responders?
  4. Rating scales are validated in Chinese populations. The references with data should be included.
  5. Demographic data didn’t seem to have included socio-economic/financial backgrounds of the students (sampling bias, confounding factor)
  6. With the limitations of data on other economic variables, the causation can’t be yet established, that the positive improvements are due to economic growth only.

Author Response

Response to Reviewer:

       We would like to express our appreciation for your extremely thoughtful suggestions. We have found your commons to be really helpful in improving the paper. We have made changes throughout the paper that address the points you have made.

  1. We agree that survey-based questionnaires have possibilities of introducing recall bias. Therefore, we have mentioned in the limitation section that “it is possible that the change trend of the studied psychological variables may stem from sampling bias and response bias”. (See line 367-368).
  2. Thank you for bringing up this issue. We are aware that the generalizability is also a limitation in our study. Thus, we have added more explanations in the context: “Additionally, the current sample is not representative of the general college population, being more resourceful, high-achiever and more urban. Our recruitment strategy thus limited the generalizability of our results”. (See line 368-371).
  3. Thank you very much for your suggestion. There was an excess of female respondents in comparison to the general student population, as over 60% of the undergraduate population at this university are females. We have run the analysis and found that there was no significant difference on age, sex, and hometown between the responders and non-responders.
  4. Thank you very much for your suggestions. We have added the Cronbach’s alpha coefficient for the scales we used in the study. (See line 156, 164 and 181).
  5. Thank you for pointing this out. In the research on college students’ mental health, ten-year data is very rare in China, and therefore has encountered many limitations. We mentioned that in the first part of the limitation section “this investigation is limited in its inclusion of sociodemographic variables due to changes in the questionnaires used throughout the years”. (See line 368-369)
  6. We cannot agree more with this point of view. The causation can’t be established without the direct measure of economic variables. Thus, findings of this study provide only preliminary evidence of the possible relationship, such that in the context of economic growth, stabilizing and improving positive mental health states could help prevent and reduce the risk of depression and suicidal ideation among college students. We also added this limitation in the manuscript. “Last, the most conspicuous problem in this study is no direct measure of economic variables. As a result, the power of discussion related to social context explanation, such as the effect of growing economy on mental health, is seriously restricted”. (See line 374-376).

Reviewer 2 Report

The authors analyzed data of cohort samples in a Chinese university and studied five mental health outcomes over a decade. As a result, the authors found that mental health improved in a decade. The observation is worth informing readers of the journal.

My concern about the manuscript is that the authors connected the mental health improvement with Chinese economic growth without reliable data. Therefore, the content of the manuscript leads to a misunderstanding of the fact. The authors might realize the misinterpretation of the conclusion because they mention lack of evidence in the 3rd comment of Limitations.
The data about students' mental health alteration in a decade is sufficient to provide to readers.

Author Response

       We would like to express our appreciation for your thoughtful suggestions. We have aware of this shortcoming and removed the possible causal statements in the abstract, discussion and conclusion part. We have also added more explanations in the limitation part. Thanks again for your thoughtful comments.

Reviewer 3 Report

Dear Authors 
Hugde interesting Study "Mental Health Outcomes among Chinese College Students over a Decade" highlights very adequately the important  relations  declining rates of suicidal ideation and depression, combined with relative stability across a range of demographic variables and also highlighted the need for more public health campaigns and interventions in universities. 
Helping students cope with mental health problems  its crucial. The research design is very appropriate, the methods adequately described. A more extended analysis of literature reveals many substantial reasons for conducting research about depression. I would kindly ask you to include article on the relationship between depression and resistance in the literature: Ortenburger D, Rodziewicz-Gruhn J, WÄ…sik J, MarfinaO, Polina N. Selected problems of the relation between pain-immunity and depression, Phys Activ Rev 2017, 5: 74-77. http://dx.doi.org/10.16926/par.2017.05.10 because the phenomenon of the coexistence of physical symptoms (including chronic pain) and depression have been widely. 

Yours faithfully, 
Reviewer

Author Response

     Thank you very much for your suggestions. We have added this reference in the literature. Thank you for your time and effort in helping us improve this manuscript. See reference 11 Ortenburger, D.E.; Bde, R.G.; Bde, J.W.; Olga Marfina, B.D.; Polina, N. Selected problems of the relation between painimmunity and depression. Physical Activity Review 2017, 5, 74-77.

Round 2

Reviewer 2 Report

The authors made appropriate corrections.